# Design of Ultra-Wideband Phased Array Applicator for Breast Cancer Hyperthermia Therapy

**DOI:** 10.3390/s23031051

**Published:** 2023-01-17

**Authors:** Cheng Lyu, Wenxing Li, Si Li, Yunlong Mao, Bin Yang

**Affiliations:** 1College of Information and Communication Engineering, Harbin Engineering University, Harbin 150000, China; 2Ocean College, Jiangsu University of Science and Technology, Zhenjiang 212003, China; 3School of Cyberspace, Hangzhou Dianzi University, Hangzhou 310018, China

**Keywords:** breast cancer, focused microwave−hyperthermia therapy (FMHT), ultra−wideband miniaturized antenna, phased array

## Abstract

Focused microwave−hyperthermia therapy has recently emerged as a key technology in the treatment of breast cancer due to non−invasive treatment. An applicator of a three−ring phased array consisting of ultra−wideband (UWB) microstrip antennas was designed for breast cancer therapy and operates at 0.915 GHz and 2.45 GHz. The proposed antenna has an ultra−wideband from 0.7 GHz to 5.5 GHz with resonant frequencies of 0.915 GHz and 2.45 GHz and dimensions of 15 × 43.5 × 1.575 mm^3^. The number of each ring was chosen to be 12 based on the SAR distribution and the performance indicators of tumor off−center focusing results for four different numbers of single−ring arrays. The homogeneous breast model is applied to a three−ring phased array consisting of 36 elements for focused simulation, and 1 cm^3^ and 2 cm^3^ tumors are placed in three different locations in the breast. The simulation results show that the proposed phased array has good performance and the capability to raise the temperature of different volumes of breast cancer above 42.5 °C after choosing a suitable operating frequency. The proposed applicator allows for precise treatment of tumors by selecting the appropriate operating frequency based on the size of the malignant tumor.

## 1. Introduction

With the increase in breast cancer patients, treatment of breast−cancer using technology has attracted wide attention. To treat breast cancer with efficacy comparable to or superior to breast−conserving surgery, several thermal techniques (extreme cold or heat) have been researched and developed. Compared to surgery and other invasive treatments such as high−intensity focused ultrasound (HIFU) [1] and cryoablation [2], thermal therapies have less scarring, better tissue preservation, superior cosmesis results, faster recovery times, and lower medical costs. Non−invasive focused microwave−hyperthermia (FMHT) is an advanced method of breast−cancer treatment that uses a variety of antennas or antenna arrays to focus the power of microwaves on human malignant tissue [3]. Breast cancer can be diagnosed by medical imaging including ultrasound [4,5], mammography [6], and MRI [7]. In addition, microwave heating shows the characteristics of selective heating because of the difference in the loss factor of each tissue [8]. The malignant tissue has a strong absorption capacity for microwaves due to the high dielectric loss factor of high−water content [9,10,11], which microwaves preferentially heat and damage instead of healthy breast tissue [12,13]. Breast−cancer microwave−hyperthermia therapy requires transducing the microwave to gather at the target and maintaining the surrounding healthy tissues below healthy temperature. Elevating the temperature over 42.5 °C at the tumor position for 60 min is the method of FMHT for breast cancer. Clinical studies have demonstrated the benefits of FMHT in breast−preservation therapy [14].

At present, most breast−cancer−treatment applicators are designed to operate on a single resonant frequency and can only accurately treat tumors of a fixed size. The applicators’ selected operating frequencies are 0.434 GHz [15,16], 0.915 GHz [17,18], 2.45 GHz [19], and higher frequencies [20]. The industrial scientific medical (ISM) band is a band mainly open to industry, science, and medicine, including 0.434 GHz, 0.915 GHz, and 2.45 GHz. The applicator working in the ISM band reduces complications and enables cost savings during device installation in the clinic [16]. Hence, the ISM band is the first choice in the selection of therapy−applicator frequency. To better treat different volumes of tumors, the proposed treatment array is designed to work at two operating frequencies 0.915 GHz and 2.45 GHz of the ISM band. Penetration at high frequencies is worse than at low frequencies [21,22,23]. Therefore, the penetration of the microwave is deeper at 0.915 GHz with a larger focus point, while there is a smaller focusing range but a lower penetration depth at 2.45 GHz [24]. After analyzing the dimensions of the tumor, the appropriate frequency could be selected. The proposed applicator has the advantage of precisely treating different tumors and protecting the health of patients.

Due to the limited size of the human breast, the designed array radius is limited. The number of antenna elements in a phased array affects the therapeutic effect. Miniaturized antennas can increase the number of elements in the phased array to treat smaller tumors due to the small dimensions. Meanwhile, it can increase the distance between the elements in the phased array and reduce the mutual coupling between antennas (<−20 dB), thus reducing excess hotspots in healthy tissue. Many types of antennas can be miniaturized to form a therapeutic phased array. The coaxial feed planar antenna is the simplest, but it has the drawback of having a huge antenna and a limited frequency range [17]. Another type of antenna is the Vivaldi antenna, but the dimensions are too large, so it has fewer elements of the phased array in the longitudinal direction and the ability to treat tumors in different quadrants of the breast is inadequate [20]. Compared to other types of antennas, microstrip antennas have several advantages, including low manufacturing costs, small dimensions, ease of manufacture, simplicity of installation, and ease of constructing a ring array. Therefore, FMHT for breast cancer uses a phased array applicator constituted by microstrip antennas.

In this paper, a three−ring phased array applicator with 3 × 12 miniaturized microstrip antenna patch antennas for breast cancer therapy was developed, utilizing the difference in frequency focus to treat tumors of different sizes. The unique contribution of this work are as follows:(1)development of oil−in−water loaded miniaturized UWB antenna to design a phased array applicator for breast−cancer therapy of different volumes of tumors;(2)selection of 0.915 GHz and 2.45 GHz (ISM band) as resonant frequencies to heat 2 cm^3^ and 1 cm^3^ tumors, respectively;(3)determine applicator dimensions and shape according to the breast dimensions and shape;(4)systematic approach for selection of the number of elements in a one−ring phased array by focusing on results;(5)verification of a three−ring phased array applicator in big numerical breast phantom for focusing power on tumors of different locations and different sizes using electromagnetic (EM) and thermal simulations.

Here is the structure of the paper: Section 2 introduces the antenna design and fabricated results; Section 3 shows the number of elements per ring selection and the three−ring phased array applicator setup and simulation results; and finally, Section 4 is the conclusion of this paper.

## 2. Antenna Configuration and Design

In this section, the design of the miniaturized antenna is introduced. The size of the radiational antenna was calculated, and the proposed antenna was optimized based on the traditional one. Meanwhile, the UWB microstrip antenna was developed to operate in the oil−water emulsion and the fabricated antennas were measured in the coupling medium.

### 2.1. Configuration of Antenna

To miniaturize a microstrip−patch antenna, slots, ground−plane deformation, and appropriate material loading are introduced. To decrease the dimension of the antenna, slots are made, and the patch’s form is modified to enhance the electric length in a compact region. Miniaturization of microstrip antennas can be achieved by fractal radiating edges, a slotted patch to prolong the current path, and ground deformation achieved by a defective ground structure. Meanwhile, another way to miniaturize the antenna is to use a substrate made of a suitable dielectric material with a high permittivity.

Figure 1 shows the designed antenna’s front, back, and side views. And Table 1 shows the dimension of the proposed antenna. The substrate of the proposed antenna utilized Rogers RT5880 with a dielectric constant of 2.2, tangent loss of 0.0009, and thickness of 1.575 mm. The dimensions of the substrate of the patch were 15 × 43.5 mm^2^. The radiating patch was composed of an ellipse−shaped microstrip patch, and all ellipses’ ratio were 0.27. The proposed antenna was fed by a feedline. Simulation results of the proposed antenna’s parameters were analyzed by Ansys Electronics 2019.

Figure 1a shows the ellipse−shaped patch with a straight feedline. Figure 1b shows the ground plane with two symmetrical quarter−circle tangents and an intermediate rectangular slot. The side view of the antenna is displayed in Figure 1c. 

The width of the feedline was calculated and optimized. The optimized width and length of the proposed antenna’s feedline were 1.1 mm and 7.5 mm, respectively. In this work, the miniaturized antenna was analyzed in a homogenous emulsion coupling medium (oil−in−water). The dielectric properties of an oil−in−water coupling medium are $\varepsilon_{r}=22.9$ and σ=0.07 [17]. The design theory created for free space radiation requires the development of a higher dielectric medium that can adjust its dielectric properties to match the tissue of interest [17]. An oil−in−water emulsion (mixture of 60% vegetable oil, 36% DI water, and 4% HLB10 surfactant) served as the coupling medium in the therapy, and the HLB10 surfactant was made up of a mixture of 46% SPAN80 and 54% TWEEN80 [17]. The emulsion coupling medium was designed for optimal microwave−power coupling between the antenna array and the breast. In addition, the skin temperature of the breast was lowered by stirring oil−in−water to ensure patients’ health during the treatment. Because the microwave energy radiated by the antenna causes the surface temperature of the antenna to overheat and the oil−water mixture to be unstable, the oil−in−water was stirred during the measurement process to cool the temperature and make the emulsion even. Considering the treated volumes of the tumors, the proposed UWB antenna operated at 0.915 GHz and 2.45 GHz in the ISM band.

### 2.2. Evolution of Antenna

The proposed antenna structure is the result of various steps in the variation of patch shape and ground shape. To comprehend the miniaturization procedure of the proposed antenna, the design evolution is indicated in Figure 2. At first, an elliptical patch with a feedline and a rectangular ground was constructed as the stricture of the initial UWB antenna. The parameters of the antenna were calculated to make it operate at the required low frequency (0.915 GHz). The current of the UWB antenna had obvious edge−distribution characteristics. The resonant frequency was formed at 0.915 GHz, but the working state was not very good at 2.45 GHz.

In step 2, two semi−elliptical slots were added in the ellipse patch to increase the high−frequency matching performance. Based on the characteristics of the current edge of the antenna, the impedance matching of the antenna could be improved by digging holes inside the antenna radiating patch or adding a special notch structure, etc. Thus, the second step of the design was adding two elliptical gaps, which lengthened the current paths and changed the excitation eigenmode, as shown in Figure 2. Although the current had a strong current−fringe effect in the microstrip antenna, the amplitude of the current in the central part of the radiating element was still very strong at low frequency. Therefore, it was necessary to keep the middle ellipse structure. As seen in Figure 3, especially at high frequency, the maximum current region was deposited at the feedline, the edges of the patch and the edge, with the slits receiving the majority of current. As shown in Figure 4, at the operating frequencies of 1.9 GHz and 2.45 GHz, the reflection coefficients were improved by 30% and 10%, respectively. However, in the process of deformation, the antenna had a notched band at 1.14 GHz–1.42 GHz, and the impedance bandwidth of the UWB antenna was unacceptable.

In the literature [24], the resonant frequency of 2.45 GHz is equally important for the treatment of breast tumors with a focal size of 2 cm^3^ or less. Thus, the shape of antenna III was developed by etching two quarter circles on either side of the ground and a rectangular slot in the center of the ground. The transformation of the ground plane improved the matching performance of the feedline, as shown in Figure 4. The current distribution of antenna III showed an increase in the current distribution on the feedline. Due to this transition, the reflection coefficient was increased to −44.8 dB at the resonant frequency 2.45 GHz. The reflection coefficients corresponding to each step of antenna evolution are shown in Figure 4.

### 2.3. Parametric Analysis

The impedance matching of the microstrip patch antenna depends on the width of the feedline to satisfy the characteristic impedance. The width of the feedline ($W_{f}$) and the radius of the inner ellipse ($L_{i}$) are the key parameters for antenna design. Thus, these parameters were optimized to achieve an efficient performance. The reflection coefficients (S11) simulated by different values of parameters are shown in Figure 5. The impedance match (50 Ω) can be achieved by adjusting the width of the feeder. As the width of the feedline decreases from 2.1 to 1.1, the higher resonant frequency decreased from 2.6 GHz to 2.45 GHz, and the lower resonant frequency increased from 0.7 GHz to 0.915 GHz, as seen in Figure 5a. More current deposited at the edge of the slit as the width of the slit increased, and the impedance match at 0.915 GHz and 2.45 GHz was enhanced. However, the higher resonant frequency shifted to 2.5 GHz when the width of the slot became bigger than 0.89 ($L_{i}$ = 10.75). Therefore, $W_{f}$ = 1.1, and $L_{i}$ = 11.25 were selected as the most appropriate parameter values for the proposed antenna.

### 2.4. Measurement Results and Discussions

The fabricated antennas were immersed in an emulsion medium (oil−in−water) composed of vegetable oil, DI water, SPAN80, and TWEEN80 to measure the reflection coefficients. The reflection coefficients of the antennas were measured by the electronic device of Agilent Technologies, E5071C ENA Series Network Analyzer, as shown in Figure 6c. Figure 7 shows the measured reflection coefficients of 3 antennas of the 36 fabricated antennas in the measured frequency band of 0.1 GHz to 5 GHz. The measurement results show that the measured reflection coefficients of the fabricated UWB antennas were below −10 dB from 0.7 GHz to 5 GHz. The lower resonant frequencies of antenna I and antenna II were 0.915 GHz. However, the higher resonant frequencies of three fabricated UWB antennas deviated from 2.45 GHz, but the reflection coefficients at 2.45 GHz were less than −15 dB and the treatment effect was not affected. This deviation may be caused by VNA cables, coupling emulsion−manufacturing tolerances, and the extended grounding effect of the SMA connecter, which were not present in the simulation part.

## 3. Selection of Phased Array Configuration

The number of antenna elements in the one−ring phased array was selected according to the effectiveness of focusing. Meanwhile, the three−ring phased array applicator was established, and tumors of different sizes were focused at three different positions of the breast phantom.

### 3.1. Ring Phased Array Design Process

The patient is lying in a prone position and the breast is inserted into the applicator filled with a coupling medium. Oil−in−water has relatively low attenuation, and the dielectric properties of the oil and water can be adjusted to match the tissue of interest by changing the ratio of oil to water [17]. To prevent excessive temperatures and oil−in−water stratification, the coupling medium is constantly agitated during the treatment. Excitations of antenna elements are controlled by power amplifiers and phase shifters, as shown in Figure 8. An optimization algorithm technique is used to determine the optimal power and phase settings of the phased array antenna to direct power deposition at the tumor position. Hence, the output power of the power amplifier is derived from the radiated power of the antenna calculated from the optimized phase and amplitude of the antenna element for selected power deposition at the breast cancer position. The internal structure information of the breast is detected by imaging detection, such as MRI, MITAT [25], and so on. During FMHT, the temperature distribution is collected and fed into the computer for focused analysis. The FMHT system can be used to treat breast cancer more accurately and avoid causing unnecessary harm to patients because the cylindrical phased array has outstanding effects in reducing focal spot size and focusing depth, according to research conducted by Chou’s group on conformal phase-controlled focusing arrays [26,27,28]. In this paper, we considered utilizing a three−ring phased array as a breast-cancer thermal applicator. Meanwhile, time reversal (TR) technology was used in this paper to optimize the phases and powers of antennas to deposit power at the tumor location. TR was used to feed the signal to the array element with phase conjugate (or reverse delay) to achieve microwave focusing [29]. The operation method of TR in EM simulation was as follows:(1)The breast model was placed in the phased array, and the dielectric properties of the breast tissue were obtained from breast imaging;(2)A dipole was placed at the tumor location within the breast;(3)the relative phases and amplitudes of antenna elements emitted from the point source (dipole) through the breast tissue and coupling liquid were calculate and stored;(4)The complex phase information was conjugated to produce the desired phase delay. In the treatment, the excitations of each antenna element were changed using a phase shifter and a power amplifier to redirect the focus to the tumor location.

The purpose of FMHT is to elevate the temperature above 42 °C in the target position while keeping surrounding normal tissue below an acceptable temperature. The temperature is calculated by Pennes bio−heat equation [30]:(1)$$C_{p}\rho \frac{\partial T}{\partial t}=\nabla \cdot \left(K\nabla T\right)+A_{0}+Q-B(T-T_{B}),$$
where $C_{p},\rho ,K,A_{0},B,T_{B},Q$ are specific heat, density, thermal conductivity, metabolic heat generation, capillary blood perfusion coefficient, blood temperature, and power dissipated per unit volume, respectively. Table 2 illustrates the thermal and dielectric properties of breast tissue. The temperature distribution of the breast model can be calculated by *Q*. The calculation equation of power density *Q* is expressed as
(2)$$Q=0.5\sigma {\left|\vec{E}\right|}^{2},$$
where Q is the density of power, σ is the conductivity, and $\vec{E}$ is the electric field [20]. Meanwhile, the SAR value is expressed by the deposited power of the electric field in the breast tissue as
(3)$$SAR=\frac{\sigma {\left|\vec{E}\right|}^{2}}{2\rho }=\frac{Q}{\rho },$$
and
(4)$$\vec{E}=\sum_{n=1}^{N}a_{n}e^{-i\varphi_{n}}{\vec{E}}_{n}.$$

In Equation (4), ${\vec{E}}_{n}$ is the electric field generated by unit power and zero−phase excitation of the nth antenna, *N* is the number of antennas, and $\varphi_{n}$ and $a_{n}$ present the input phases and amplitudes of array elements, respectively. Focused treatment is improved by optimizing the phases and amplitudes of antenna elements [31]. By analyzing the SAR distribution, the focusing effect is verified, and then the excitation of antenna element is optimized. Meanwhile, the temperature in Equation (1) can be obtained from the SAR value calculated in Equation (3). Therefore, the SAR distribution and the temperature distribution are observed to determine the effect of focusing on EM simulation and thermal simulation, respectively.

Tumors have a higher electric−loss factor than other tissues [32,33], indicating that power is more concentrated on the tumor to achieve the purpose of FMHT. The control of tumor−location temperature is simplified to optimize the focusing effect of SAR distribution. Therefore, several treatment indicators have been designed to determine the ability to focus by analyzing the SAR value. The average power−absorption ratio (APA) [16], hotspot−to−target quotient (HTQ) [16], and tumor coverage of 50% (TC_50_) are the indicators used to judge the effectiveness of the treatment and are expressed as
(5)$$aPA=\frac{{\bar{P}}_{t\arg et}}{{\bar{P}}_{healthy\_tissue}},$$
(6)$$HTQ=\frac{{\bar{SAR}}_{V1}}{{\bar{SAR}}_{t\arg et}},$$
and
(7)$$TC_{50}=\frac{V_{t\arg et}(SAR>\max(SAR)/2)}{V_{tumor}},$$
where ${\bar{P}}_{t\arg et}$ and ${\bar{P}}_{healthy\_tissue}$ are the average power absorbed in the desired tumor position and healthy tissues, And *V*1 is the 1% of the healthy tissue volume with the highest amount of specific absorption rate (SAR). In Equation (6), ${\bar{SAR}}_{V1}$ and ${\bar{SAR}}_{t\arg et}$ are defined as the average SAR in the V1 and the target, respectively. Maximizing the aPA value and minimizing the HTQ value means accumulating more power in the target area, and reducing hotspots in healthy tissue to achieve better treatment. Tumor coverage TC_50_ is defined as the proportion of tumor volume covered by 50% of the maximum specific absorption (SAR) [34]. The larger the TC_50_, the better the treatment effect.

EM modeling utilized Ansys Electronics 2019 for simulation. The skin surface was exposed to 20 °C circulating oil−in−water in the simulation to reduce skin−surface temperature during treatment. The chest and breast were set at 37 °C. The pulse time was thermally simulated for 60 min to simulate the high temperature. Power was output to each antenna until it reached the target region of 45 °C.

### 3.2. Optimization Number of Elements

The elements were separated by less than half a wavelength and coupled with each other to produce excess focal points that returned to the patient’s healthy tissue. After optimizing the number of elements under the condition of fixed array radius, the distance between the antennas was rational and the influence of mutual coupling on the therapy was weakened, which helped to decrease the production of additional undesirable hot spots brought by focusing and protected the patient’s healthy tissue from being burned. The average radius of the breast for middle−aged women is from 50 mm to 70 mm [35]. Hence, the radius of the homogeneous breast phantom for the single−layer phased array was set as 65 mm. The radius of the ring−phased array was set to 100 mm to accommodate most breast sizes. The proposed miniaturized UWB antenna was used to compose the circular phased array, which increased the number of elements and reduced the focusing size [36]. The oil−in−water was filled in the phased array as a coupling medium. The homogeneous breast model (including chest, skin, fat, and tumor) was immersed in an emulsion medium for the simulation to determine the focusing effect of the phased array applicator. The 1 cm^3^ and 2 cm^3^ tumors were set in the breast model, the thickness of the skin was 1 mm, and the dimensions of the fat were 65 × 65 × 104 mm^3^. The effect of the number of elements on the focusing effect was verified by the focusing simulation results of single-ring arrays containing different numbers of elements, as shown in Figure 9.

When a focusing target was off−center, it was more difficult to focus than when target was located in the center of the breast. Therefore, the focusing performance was analyzed when the tumor was placed 30 mm from the center. The height of the emulsion container was 120 mm. The antenna elements were distributed in the container and a ring−phased array was equally formed 100 mm from the center of the tank. The numbers of elements per ring were defined as 8, 10, 12, and 14. The phased arrays used TR to focus the microwave at the desired position. The normalized SAR−distribution results for different one−ring phased array applicators are shown in Figure 10.

From Figure 10, it is clear that the focusing range of 0.915 GHz was larger than 2.45 GHz. Therefore, the UWB phased array applicator could accurately treat tumors of different sizes. As shown in Figure 10a–c, the focus position and burning range were more accurate as the number of units increased from 8 to 14 at the resonant frequency of 0.915 GHz. In the vertical view, a focal length was obviously greater than 20 mm, which could be solved by increasing the number of layers of the phased array. The focal range was 1 cm^3^ at the resonant frequency of 2.45 GHz, shown in Figure 10e–h. When there were 8 elements in a one−ring phased array, the skin could receive enough microwave power to cause injury due to poor penetration of 2.45 GHz, as shown in Figure 10e. Many unwanted hotspots appeared in healthy tissue due to coupling caused by the close distance between the antennas when there were 14 elements in a one−ring phased array operating at 2.45 GHz (Figure 10h). The SAR distributions of two resonant frequencies illustrated that 12 elements per ring had the best focusing effect compared to other numbers and had the least damage to surrounding healthy tissue at both working frequencies.

Maximization of aPA and minimization of HTQ represent good hyperthermia effects. In other words, the more different they are in the bar diagram, the better the focus effectiveness, as shown in Figure 11. The largest difference between aPA and HTQ occurred when the number of elements in the one−ring phased array was 12 at two resonant frequencies. From Figure 11a, it is illustrated that the aPA rose above 11 and the value of HTQ was close to 1 by increasing the number of antennas to 12, and there was not much difference between 12 and 14. Meanwhile, when the operating frequency was 2.45 GHz, the aPA value was a maximum of 13.06 and the HTQ value was a minimum of 0.705 when the number of antennas was 12, as shown in Figure 11b. When the number was 12 at 2.45 GHz, the HTQ value was reduced by an average of 29%, and the aPA value increased by an average of 18.3% compared with other cases, respectively. Thus, 12 is the appropriate number of elements for a one−ring phased array design. The number of elements in a three−ring phased array applicator was determined to be 3 × 12.

### 3.3. Three−Ring Phased Array Simulation

About 62% of breast cancers occur in the upper outer quadrant of the breast, with the rest occurring in the middle and lower quadrants of the breast [16]. Therefore, in order to verify the treatment effectiveness of the three−ring phased array applicator, three tumor locations are selected, the center, upper outer quadrant, and lower quadrant of the breast, as shown in Figure 12a,b. To verify that the three−ring phased array applicator could accurately treat breast cancer in different locations, a longer breast model was used, and three breast locations where cancer frequently occurs were set as the focal points. The long breast model consisted of skin with a thickness of 1 mm, a chest wall with a volume of 240 mm × 240 mm × 20 mm, and a semi−elliptic spherical adipose tissue with a radius of 65 mm and a height of 130 mm. Tumors of 1 cm^3^ and 2 cm^3^ were placed in the three cancer-generating locations, and the larger tumors were treated using 0.915 GHz and the smaller tumors were treated using 2.45 GHz, depending on the difference in the focal range according to two resonant frequencies. The UWB three−ring phased array applicator was composed of the proposed miniaturized UWB microstrip antennas for focusing tumors of different sizes at 0.915 GHz and 2.45 GHz. To evaluate its applicability in biomedical applications and to ensure the safety of using the proposed phased array applicator, the longer breast model was used for focusing simulation and the oil−in−water was the coupling medium covering the phased array, as shown in Figure 12c,d. The relative position of the elements was the same in the bottom and the top ring arrays, and the elements of the second layer were rotated by 15° relative to the other two layers. The phased array radius was 100 mm. The distance between the array elements on the same layer was 52.7 mm, and the vertical distance between the tops of adjacent layers array elements was 47.5 mm, as shown in Figure 12c. The fabricated three−ring phased array applicator is shown in Figure 13.

As shown in Figure 14a,b, the output reflection coefficients $S_{nn}(n=1,\cdots ,36)$ of 36 elements of the phased array were below −10 dB in the case of no breast model and with the breast model at the desired operating frequencies. The S−parameter measurements showed that the antenna elements performed better at 0.915 GHz and 2.45 GHz. In addition, $S_{mn}(\begin{array}{cc} m,n=1,\cdots ,36. & m>n \end{array})$ represents the mutual coupling between antennas, where the coupling between adjacent elements is relatively large. As shown in Figure 14b, the S_mn_ of the phased array was less than −23 dB at two resonant frequencies without the breast model. When the phased array was simulated with the breast model, the coupling coefficients were below −18 dB at 0.915 GHz and below −21 dB at 2.45 GHz, as shown in Figure 14d. For use as a breast−cancer treatment applicator, the structure of the phased array was reasonable, and the distance between antenna elements was appropriate.

To verify the focus treatment capability of the designed phased array, Figure 15 and Figure 16 show the focusing effects of 1 cm^3^ and 2 cm^3^ tumors at different resonant frequencies of 0.915 and 2.45 GHz, respectively, where the focusing effect is demonstrated by the steady−state temperature and SAR distributions. Due to the small focusing range at high frequency, 2.45 GHz was used to focus on a 1 cm^3^ tumor, while 0.915 GHz was utilized to treat a 2 cm^3^ tumor. The distance between the antenna elements was less than half a wavelength and antennas radiated in the near field, thus coupling between antennas affected the focusing effect. As shown in Figure 15 and Figure 16, the microwave was focused on the desired positions, and no focus shift occurred in all cases. From the SAR distributions, the operating frequency of 0.915 GHz produced more redundant energy near the tumors compared to 2.45 GHz. However, the energy was much lower than that of malignant tumors, so it could not harm the surrounding healthy tissues, as shown in the temperature distribution of Figure 16. When the tumor was close to the skin, the SAR and temperature distribution at 2.45 GHz had a better focusing effect than at 0.915 GHz. The temperature distribution of the focus point at 2.45 GHz was much tighter than that at 0.915 GHz. The simulation results illustrate that the UWB applicator had high thermal treatment performance at both resonant frequencies.

The indicators of aPA ratio and HTQ were calculated to measure the treatment performance of the proposed phased array applicator. Figure 17 shows the values of the aPA ratio, HTQ, and TC_50_ of the three−ring phased array applicator operating at 0.915 GHz and 2.45 GHz with different volumes of tumors in different quadrants of the breast model. As can be seen in Figure 17a, the HTQ values for all cases were below 0.7. Due to low penetration at 2.45 GHz, the value of HTQ was at least 0.172 larger and the aPA value was at least 5.89 lower than the other two cases when the tumor was located at the center of the breast. The focusing performance of 0.915 GHz was much better for tumors deep in the breast compared to the 2.45 GHz case. When the tumor was located closer to the skin, the aPA ratio was above 15 for both resonant frequencies. When the tumor was located on the low quadrant of the breast, the aPA value of 2.45 GHz reached 20. Finally, the percentage of TC_50_ at the two resonant frequencies was above 95% in all three cases. The TC_50_ value was close to 100% when the tumor was located in the center of the breast, as shown in Figure 17c.

The accuracy of treatment is critical for breast−cancer−focusing microwave−hyperthermia therapy. The distance between antennas is less than half a wavelength and the antennas of phased arrays radiate microwaves in the near−field. The coupling between antennas affects the radiate efficiency of the proposed antenna and may cause deviations in resonant frequencies and create excess hot spots when focusing. Therefore, the proposed miniaturized UWB microstrip antenna in this paper is designed to solve the above problems. The antenna size is positively correlated with the operating frequency, and the proposed antenna is smaller in size than the single−antenna applicators used for breast cancer. In addition to the design of the antenna structure, the coupling medium environment in which the antenna works also contributes to the miniaturization of the antenna. The proposed three−ring phased array applicator is established by the UWB antenna operating at 0.915 GHz and 2.45 GHz to treat 1 cm^3^ and 2 cm^3^ tumors, respectively. Compared with the narrow−band antenna forming the 0.915 GHz phased array breast−cancer applicator [17], the UWB antenna is smaller in size (15 × 43.5 mm^2^), especially in width. Due to the proposed miniaturized antenna, the distance between antennas is stretched, and the unwanted hotspots at 0.915 GHz are weakened. Meanwhile, because the two resonant frequencies (0.915 GHz and 2.45 GHz) of the proposed antenna belong to the ISM band, there are no additional regulatory complications and costs associated with the clinical installation of the applicator. From Table 3, it is clear that the breast−cancer applicators reported in [20,34] have high installation costs. In the literature [34], the multi−resonant applicator is composed of four antennas of different sizes operating at 1.5 GHz, 2.5 GHz, 3.5 GHz, and 4.5 GHz to treat a 3 cm^3^ tumor in the upper outer quadrant of the breast, a 2 cm^3^ tumor in the center of the breast, and a 1 cm^3^ tumor in the lower quadrant of the breast. However, when changing the size of the tumor in the original breast position, it is necessary to rearrange the positions of different sizes of antennas to form a phased array to obtain better treatment results. However, instead of rearranging the applicator, the proposed UWB phased array just needs to be adjusted to work at the suitable frequency. The emulsion coupling covering the applicator is necessary to reduce the temperature to protect the patient. In the literature [18], the applicator operates at 0.915 GHz and the simulation results show that the focus range is relatively large and unwanted hotspots occur in healthy tissue. Moreover, the aPA ratio is below 1.5 when the target is close to the chest wall. Compared with other applicators for the treatment of breast cancer, the proposed applicator has a lower HTQ and higher aPA ratio. The proposed phased array applicator could accurately therapy tumors of different volumes by selecting a suitable working frequency and could have a high treatment effect.

## 4. Conclusions

In this paper, a UWB three−ring phased array applicator consisting of the proposed ultra−wideband microstrip antennas was developed for breast cancer thermal therapy. The proposed antenna has an ultra−wideband with two resonant frequencies of 0.915 GHz and 2.45 GHz and a volume of 15 × 43.5 × 1.575 mm^3^. Through simulating one−ring phased arrays with different numbers of elements, the appropriate number of elements for two operating frequencies was found to be 12, depending on the performance of the FMHT. In contrast to other applicators that obtain a single resonance frequency, the three−ring phased array contains 36 elements radiating at two resonant frequencies in the ISM band. Time reversal is used to optimize the excitations of antennas so that the microwaves are concentrated at the desired tumor location. The simulation results show that the designed applicator is effective at 0.915 GHz and 2.45 GHz for tumors of 1 cm^3^ and 2 cm^3^ at three different locations, respectively. Therefore, the proposed applicator that treats the tumor at the appropriate resonant frequency is effective by analyzing the tumor volume. Further research should be conducted to implement global optimization algorithms to optimize the excitations of antenna elements and to use real breast models to improve the accuracy of treatment.

## Figures and Tables

**Figure 1 sensors-23-01051-f001:**
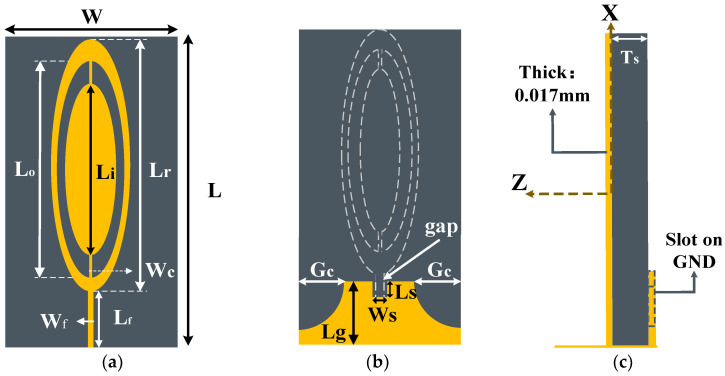
Geometries of the proposed antenna. (**a**) Front view. (**b**) Back view. (**c**) Side view.

**Figure 2 sensors-23-01051-f002:**
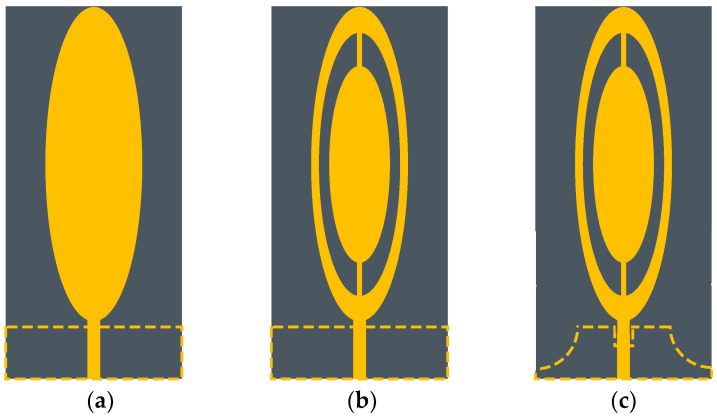
Evolution steps of ellipse-monopole-antenna changes: (**a**) antenna I, (**b**) antenna II, (**c**) antenna III.

**Figure 3 sensors-23-01051-f003:**
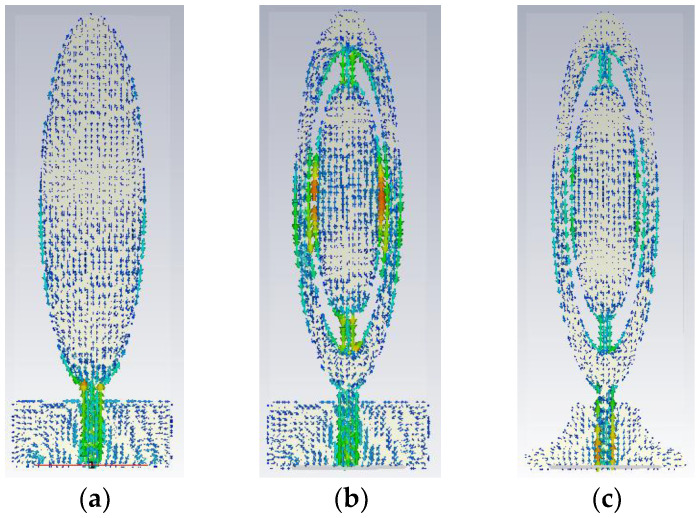
The surface current distribution of the proposed antenna in three stages at 2.45 GHz: (**a**) antenna I, (**b**) antenna II, (**c**) antenna III.

**Figure 4 sensors-23-01051-f004:**
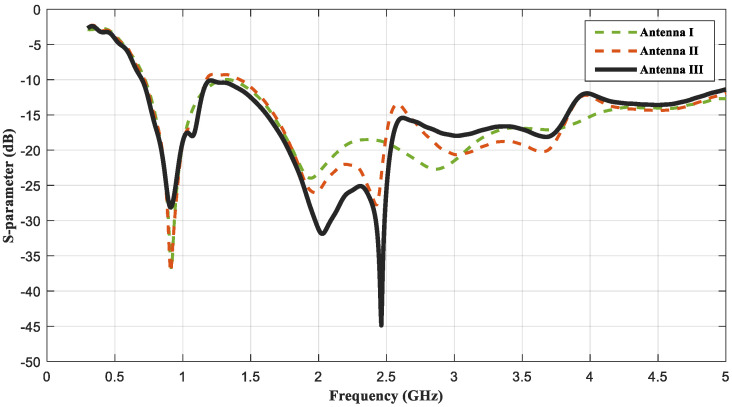
Comparison of the reflection coefficient of three evolution stages.

**Figure 5 sensors-23-01051-f005:**
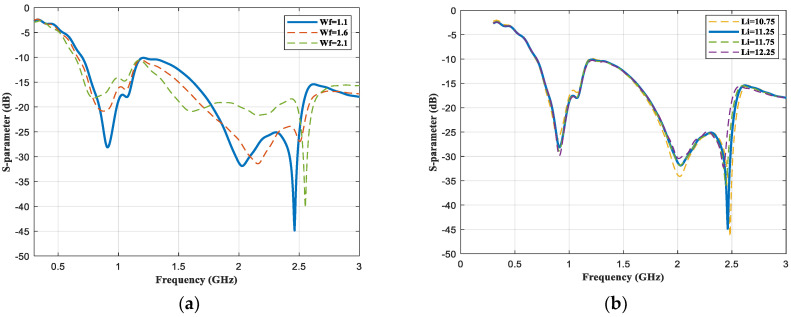
Comparison of the reflection coefficient of different values of $W_{f}$ and $L_{i}$, respectively. (**a**) S_11_ of different $W_{f}$ and (**b**) S_11_ of different $L_{i}$.

**Figure 6 sensors-23-01051-f006:**
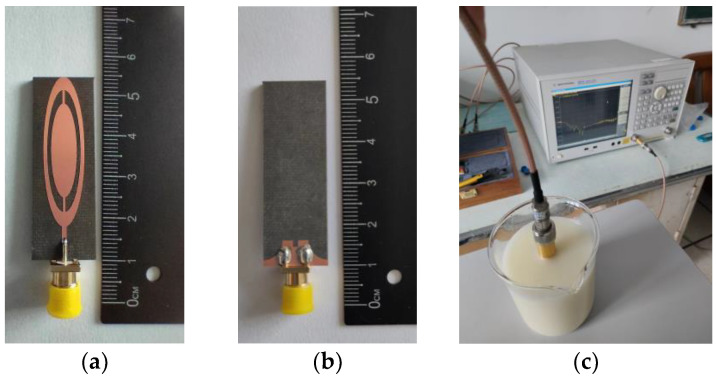
The dielectric properties of the emulsion−coupling medium were measured using an open−ended coaxial dielectric probe kit. (**a**) The front view of the fabricated antenna, (**b**) the back view of the fabricated antenna, and (**c**) the measurement of the fabricated antenna.

**Figure 7 sensors-23-01051-f007:**
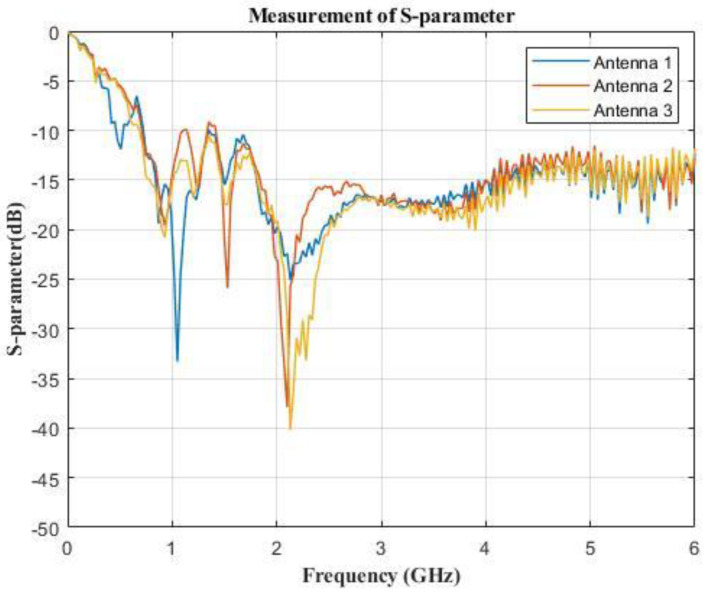
The reflection coefficient measurement of four antennas.

**Figure 8 sensors-23-01051-f008:**
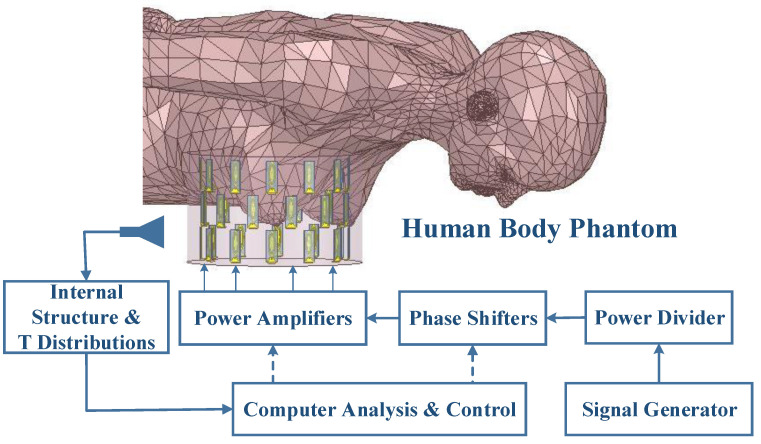
The realistic experimental setup of the FMHT modality.

**Figure 9 sensors-23-01051-f009:**
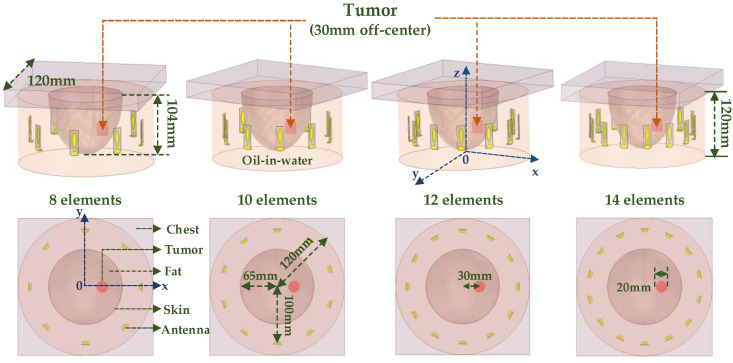
The layout of one−ring phased array configurations with a different number of antennas containing 8, 10, 12, and 14 elements, separately. The breast phantom includes chest, fat, skin, and tumor (30 mm away from the center).

**Figure 10 sensors-23-01051-f010:**
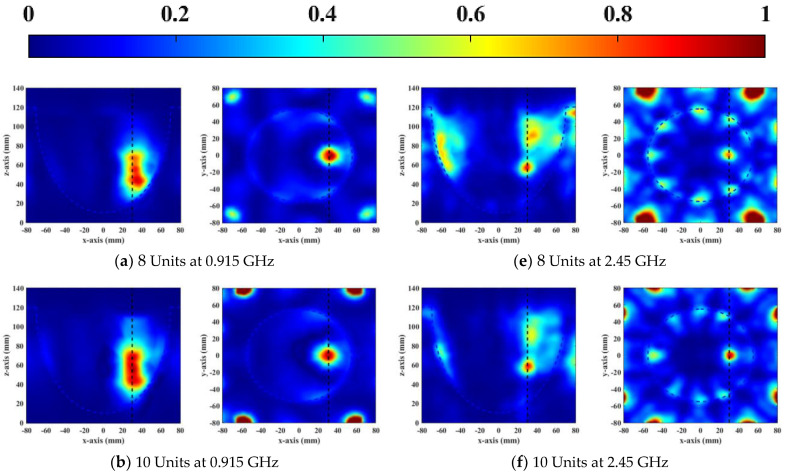
Normalized SAR distribution (W/kg) of vertical (XY plane) and horizontal (YZ plane) cross−sectional views of one−ring phased array containing 8, 10, 12, and 14 elements within an emulsion−coupling medium at 0.915 GHz (**a**–**d**) and 2.45 GHz (**e**–**h**), respectively.

**Figure 11 sensors-23-01051-f011:**
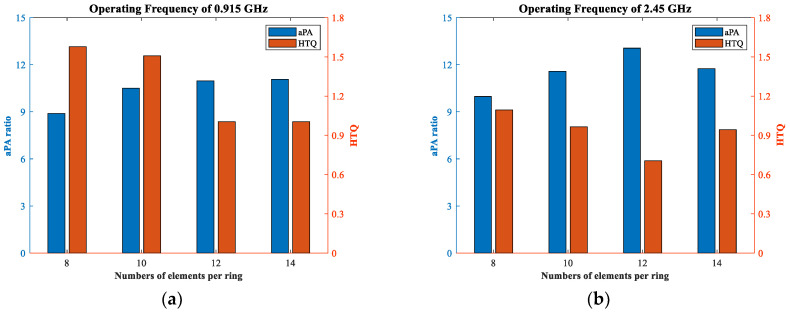
Influence of the number of antennas in one ring. aPA ratio and HTQ of four different numbers of antennas in one−ring phased array at 0.915 GHz and 2.45 GHz, respectively. (**a**) aPA ratio and HTQ of four one−ring phased arrays at 0.915 GHz. (**b**) aPA ratio and HTQ four one−ring phased arrays of 2.45 GHz.

**Figure 12 sensors-23-01051-f012:**
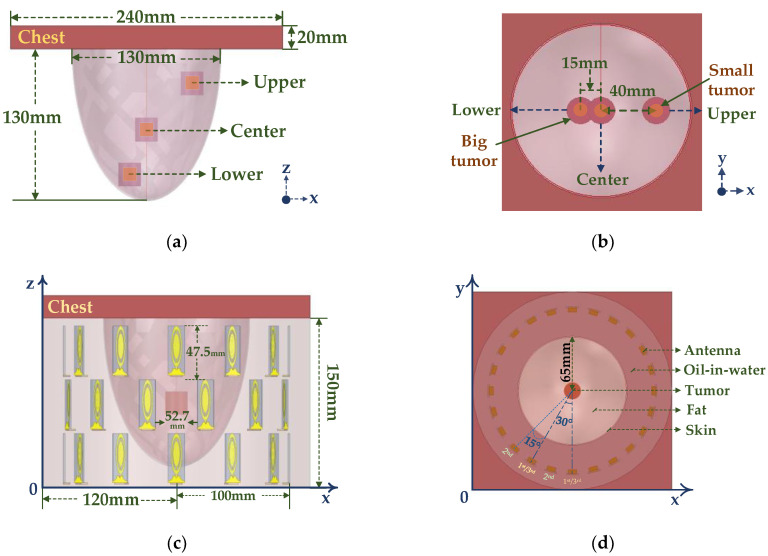
Illustration of the homogeneous breast model with tumors in three different locations and the proposed UWB phased array applicator. (**a**,**b**) Horizontal (XZ plane) and vertical (XY plane) views of the breast model (XY plane); (**c**,**d**) Horizontal (XZ plane) and vertical (XY plane) views of the three−ring phased array applicator containing a breast model with a tumor in the center of the breast.

**Figure 13 sensors-23-01051-f013:**
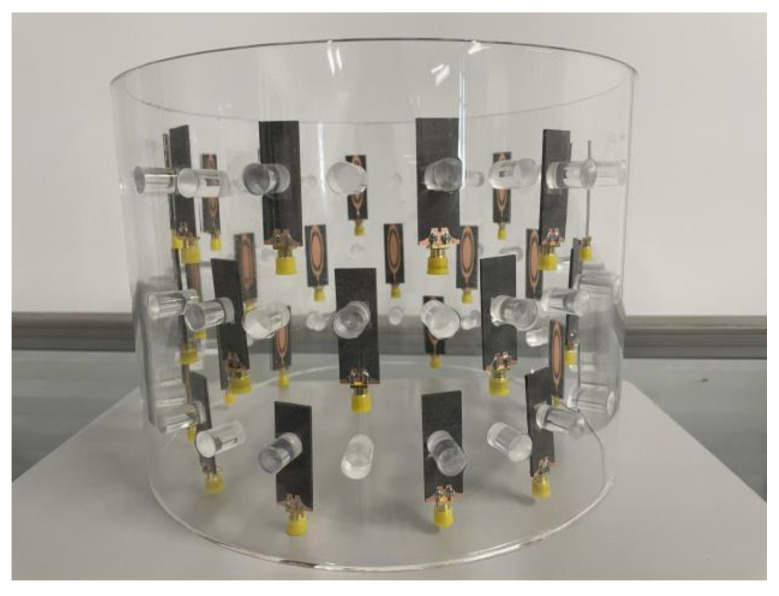
Side view of fabricated three−ring phased array applicator.

**Figure 14 sensors-23-01051-f014:**
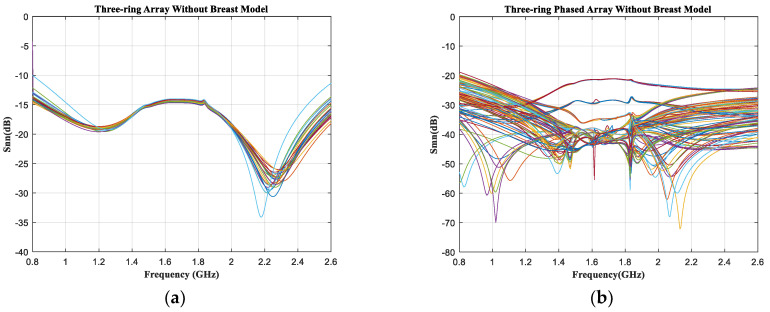
Simulation results of S parameters of 36 elements in the three−ring antenna array. (**a**,**c**) Reflection coefficient (S_nn_) of the three−ring array without the breast model and with the breast model immersed in oil−in−water. (**b**,**d**) Mutual coupling (S_mn_) of two cases, separately.

**Figure 15 sensors-23-01051-f015:**
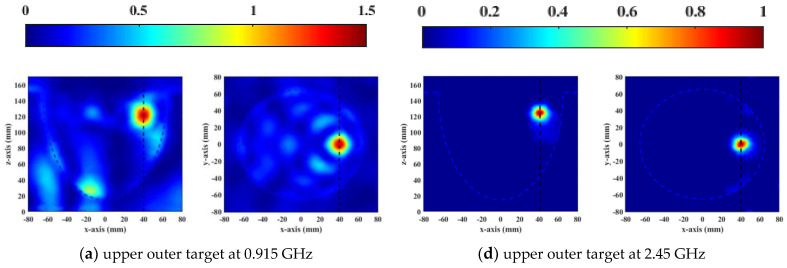
Vertical (XY plane) and horizontal (YZ plane) cross−sectional views of SAR distribution (W/kg) for time−reversal focusing at the upper outer, center, and lower quadrants of the breast at 0.915 GHz and 2.45 GHz, respectively.

**Figure 16 sensors-23-01051-f016:**
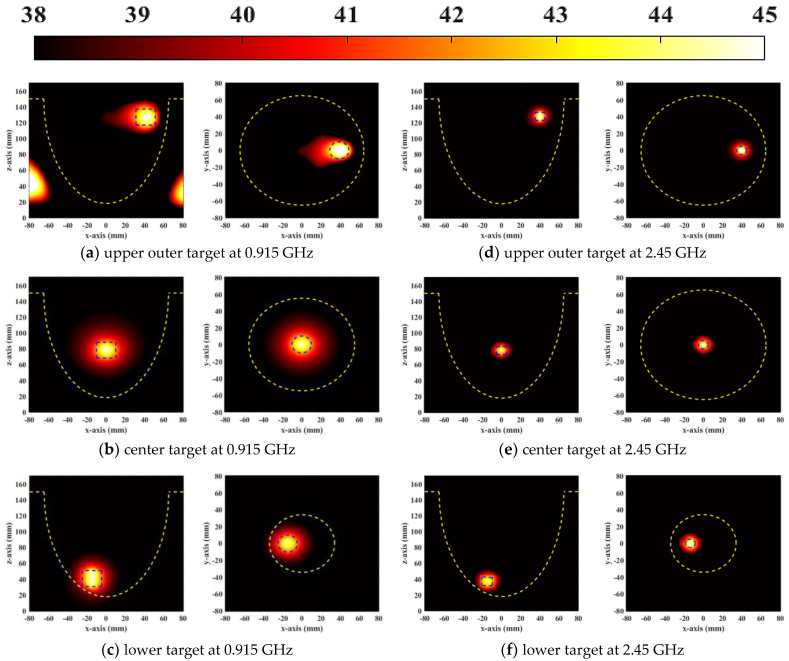
Vertical (XY plane) and horizontal (YZ plane) cross−sectional views of temperature distribution (°C) for long breast phantom with 1 cm^3^ (0.915 GHz) and 2 cm^3^ (2.45 GHz) tumor at three different locations.

**Figure 17 sensors-23-01051-f017:**
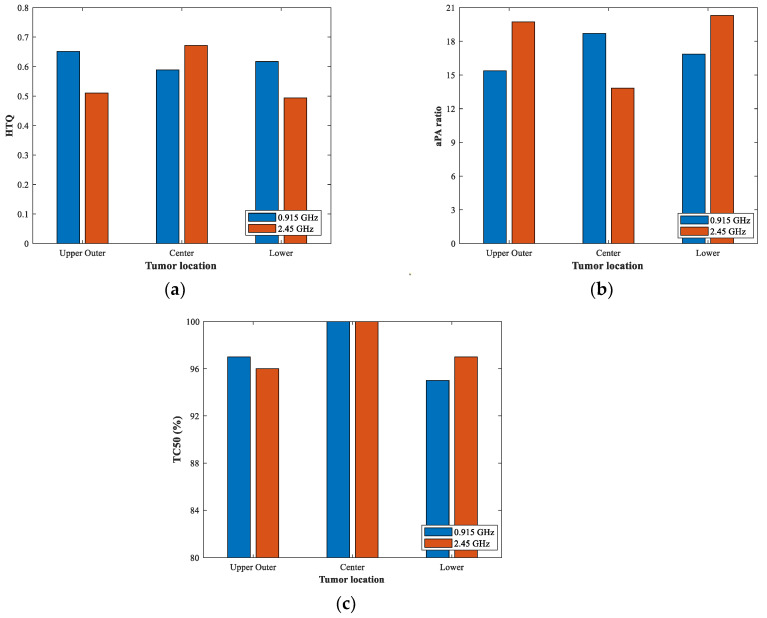
(**a**–**c**) Performance configurations compared between different frequencies considering different tumor dimensions and positions.

**Table 1 sensors-23-01051-t001:** Dimension of the proposed antenna (Unit: mm).

Variable	Value	Variable	Value	Variable	Value
W	15	L	43.5	L_f_	7.5
W_f_	1.1	L_r_	36	L_g_	6
W_c_	0.56	L_o_	29.35	T_s_	1.575
W_s_	1.18	L_i_	22.5	Thick	0.017
G_c_	4.5	L_s_	3	gap	1.5

**Table 2 sensors-23-01051-t002:** Thermal and Dielectric Properties of Tissues in Numerical Breast Phantoms [20].

Tissue	*C_P_* (J/kg°C)	*ρ* (kg/m^3^)	*K* (W/m°C)	*A*_0_ (W/m^3^)	*B* (1/s)	*ε_r_*	σ (s/m)
Chest	3421	1041	0.49	1046	0.003	56.86	0.80
Skin	3391	1085	0.37	1620	0.002	50	1.0
Fat	2279	1069	0.306	350	0.001	15	0.15
Grand and Tumor	3600	1050	0.5	390	0.003	45	0.5

**Table 3 sensors-23-01051-t003:** Comparison of the Proposed Phased Array Breast Applicator with Prototypes Reported in the Literature.

Reference	Substrate Material	Antenna Size (mm^3^)	Array Size (mm^3^)	Frequency (GHz)	Coupling Medium	Number of Units	aPA Ratio	HTQ
[37]	FR−4	42.46 × 48.46	--	2.885	TS	1	NQ	NQ
[38]	Jeans	28 × 30	--	3, 6	NM	1	NQ	NQ
[35]	Kapton Polyimide	13 × 13	--	8, 12.4	TS	1	NQ	NQ
[39]	NQ	30 × 35	--	2.8	Castor Oil	1	NQ	NQ
[20]	Rogers RT6010	450 × 450 (approx.)	NQ	4.2	NM	24	NQ	NQ
[16]	Roger RO3010	36 × 38	131 × 131 × 161	0.434	DI water	18	3.93–20.82	0.56–1.34
[17]	FR−4	39.3 × 39.3 (approx.)	75 × 75 × NM	0.915	Oil−in−water	36	NQ	NQ
[34]	Kapton Polyimide	10^2^, 20^2^, 30^2^, 40^2^	NQ	1.5, 2.5, 3.5, 4.5	NM	35	19	NQ
[24]	Arlon AD1000	20 × 30	80 × 80 × 120	2.45	Castor Oil	24	NQ	NQ
[18]	Kapton Polyimide	NM	115 × 115 × 115	0.915	DI water	16	1.5–5	NQ
This work	Rogers RT5880	15 × 43.5	100 × 100 × 150	0.915/2.45	Oil−in−water	36	13.84–20.3	0.51–0.672

NM—Not Mention; NQ—Not Quantified.

## Data Availability

Not applicable.

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
