# Peer review of "Design of Ultra-Wideband Phased Array Applicator for Breast Cancer Hyperthermia Therapy"

_sensors, 2023, doi:10.3390/s23031051_

Round 1

Reviewer 1 Report

In this manuscript, an applicator of a three-ring phased array of ultra-wideband microstrip antenna is designed for breast cancer therapy. The research topic is interesting and timely required. The following comments could help improving the quality of the manuscript:

1. The literature review part should be removed from the Introduction section and assigned as separate section.  The literature review section should provide a general overview about the methods used for breast cancer therapy.

2. A list of contributions should be added to the Introduction section to highlight the significance of the proposed study.

3. Some figures/Table captions need to be improved. Captions should be self-explanatory (for example: Figure 5 and Table 2)

4. Figures 7 and 14 need a legend.

5. The size and resolution of the figures that include curves need to be improved.

6. The manuscript needs to be reviewed by a native speaker for English editing.

Author Response

Thank you for your precious comments and advice. Those comments are all valuable and very helpful for revising and improving our paper, as well as the important guiding significance to our research. We have studied the comments carefully and have made corrections which we hope meet with approval. Revised portions are marked in red on the paper. The main corrections in the paper and the responses to the reviewer’s comments are as flowing:

Q1: The literature review part should be removed from the Introduction section and assigned as separate section.  The literature review section should provide a general overview about the methods used for breast cancer therapy.

Thank you for pointing out this problem in the manuscript. In our revisions, we have removed the literature from the Introduction section.

Q2: A list of contributions should be added to the Introduction section to highlight the significance of the proposed study.

Thank you for the above suggestion. We have added a list of unique contributions in the Introduction section in lines 81-92. To avoid repetition, we have changed the sentence before the list.

Q3: Some figures/Table captions need to be improved. Captions should be self-explanatory (for example: Figure 5 and Table 2).

Thank you so much for your careful check. We have changed the caption of Figure 5 to ‘Comparison of the reflection coefficient of different values of Wf and Li, respectively.’ (‘Effect of variation of Wf and Li’ in the original manuscript). And the caption of Table 2 has changed to ‘Thermal and Dielectric Values of Tissues in Numerical Breast Phantoms’ (‘Thermal and Dielectric Properties of Breast Tissues’ in the original manuscript).

Q4: Figures 7 and 14 need a legend.

Thank you for your rigorous advice. We have added the legend to Figure 7. In most cases, adding legends to the S-parameter and mutual coupling coefficient plots can help readers understand the data. The Snn value is less than -10dB on the resonant frequencies and the Smn value is less than -20dB on the working frequencies, which indicates that the designed phased array applicator has good performance. Therefore, Figure 14 is used to show that the performance of both reflection coefficients of antenna elements and mutual coupling coefficients between array antennas can meet the requirements when the phased array is focused in the coupling medium and the coupling medium with the breast model inside. However, Figure 14 shows the Snn and Smn of a 36-element phased array, which means that Figures 14 (a) and (c) have 36 polylines and more than 300 polylines are shown in Figures 14 (b) and (d). We have added the legend as you mentioned, but because the legend is too long to block the data, we gave up the change and explained Figure 14 in the paper. Many thanks for your kind help!

Q5: The size and resolution of the figures that include curves need to be improved.

Thank you for the above suggestion. We have increased the font size in Figures 1,8, and 12 and expanded the size of the image. And Figures 2, 3, 4,7, 9, 11, 13, 14, and 17 have increased the size of the image. The data in Figures 10,15 and 16 is produced by importing simulation results from Electromagnetic simulation software into Mathematical software to make the pictures look better. We have increased the output step size of simulation results in Electromagnetic simulation software to clarify the pictures.

Q6: The manuscript needs to be reviewed by a native speaker for English editing.

We are very sorry for the mistakes in this manuscript and inconvenience they caused in your reading. The manuscript has been thoroughly revised and edited by a native speaker, so we hope it can meet the journal's standards. Thanks so much for your useful comments.

Reviewer 2 Report

The paper presents a system composed of ring antenna arrays with the aim of obtaining a near-field focusing system for the treatment of breast cancers via hyperthermia. From what has been shown, the work is based on simulations/optimizations performed using the electromagnetic simulator Ansys Electronics 2019. The spirit of the paper proves to be beneficial in advancing the fight against cancer. However, there are some things that should be improved in order to make the paper more intelligible. Below are the comments:

1. The font of some figures (especially figures 10, 15 and 16) needs to be enlarged because they are difficult to read. Moreover, I suggest a check of figure 8.

2.      When it is said, on line 374, that the distance between radiators of different layers is 47.5 mm, is it meant that the distance between the phase centres of the antennas is 47.5 mm?

3.      The implementation of time reversal technology needs to be explained better.

4.      In lines 263-268 there seem to be some redundant terms (for example, Cp is reported twice).

5.      I would recommend changing the sentence "The relation between the temperature and the power density is linear [11]:" on line 257. This is because this sentence is not directly inherent in equation (1) which relates Q to the squared magnitude of the electric field.

6.      How the equations (1)-(5) are implemented should be better explained. Also, equations (3)-(5) would need references.

7.      I would suggest checking the English of the paper a bit.

8.      In the introduction, I would suggest including some references to the dielectric characterization of breast tissues and some references in the field of breast cancer diagnostics. This is to give a better picture of the situation in the fight against breast cancer.

Author Response

Thank you for your precious comments and advice. Those comments are all valuable and very helpful for revising and improving our paper, as well as the important guiding significance to our research. We have studied the comments carefully and have made corrections which we hope meet with approval. Revised portions are marked in red on the paper. The main corrections in the paper and the responses to the reviewer’s comments are as flowing:

Q1: The font of some figures (especially figures 10, 15, and 16) needs to be enlarged because they are difficult to read. Moreover, I suggest a check of figure 8.

We feel sorry for the inconvenience brought to the reviewer. We have increased the font size of the axials and increased the clarity of the picture in figures 10, 15, and 16. And thank you for pointing out the problem in figure 8. We have reviewed and modified the process of FMHT in figure 8. The internal structure of the breast is collected and temperature distribution is collected by imaging detection method during the treatment, so ‘Q & T’ in the module was changed to ‘Internal Structure & T Distribution’. And ‘Excitation Optimization’ refers to the optimization of excitations of elements in the phased array in the optimization process, so it was not suitable to use it to explain the system setting and we have changed it to ‘Computer analysis & control’. The collected information is fed into a computer for analysis and then optimized excitations are fed back to power amplifiers and phase shifters. Meanwhile, we have changed the interpretation of the image below figure 8.

Q2: When it is said, on line 374, that the distance between radiators of different layers is 47.5 mm, is it meant that the distance between the phase centers of the antennas is 47.5 mm?

We feel sorry for the inconvenience brought to the reviewer. The vertical distance between the tops of antennas with adjacent layers is 47.5mm, in other words, the ring spacing plus the antenna length is 47.5mm. We have shown it in figure 12 and changed the interpretation of 47.55mm.

Q3: The implementation of time reversal technology needs to be explained better.

We are grateful for the suggestion. To clarify and by the reviewer’s concerns, we have added a more detailed interpretation regarding time reversal technology. More detailed statistical analysis was added on page 8 and line 238-250. We have shown the workflow of time reversal technology in EM simulation.

Q4: In lines 263-268 there seem to be some redundant terms (for example, Cp is reported twice).

Thank you for pointing out this problem in the manuscript. In our revisions, we have removed redundant interpretation of the data in Table 2 from the paper.

Q5: I would recommend changing the sentence "The relation between the temperature and the power density is linear [11]:" on line 257. This is because this sentence is not directly inherent in equation (1) which relates Q to the squared magnitude of the electric field.

Thank you for pointing out this problem in the manuscript and we deeply appreciate the reviewer’s suggestion. Q does not have a direct linear relationship with temperature. The following formula is the calculation formula of energy power Q and the electric field E. Therefore, we have changed ‘The relation between the temperature and the power density is linear [11]’ to ‘The calculation formula of power density Q is expressed as’.

Q6: How the equations (1)-(5) are implemented should be better explained. Also, equations (3)-(5) would need references.

Our deepest gratitude goes to you for your careful work and thoughtful suggestions that have helped improve this paper substantially. We have changed the position of the temperature (T) formula (equation 1) and energy (Q) formula (equation 2) to explain the relationship between formulas more clearly. And we have added the calculation formula of SAR (equation 3) and showed the relationship between SAR and Q. Moreover, the formula of the total electric field obtained from the amplitude and phase of the antenna has been added to the paper (equation 4). Through optimization of excitations of antenna elements, SAR distribution shows the power deposit on the desired location, and then the T distribution is calculated accordingly. We have added a detailed description of equations (5-7) in lines 278-285. Each equation has been labeled with references.

Q7:  I would suggest checking the English of the paper a bit.

We are very sorry for the mistakes in this manuscript and inconvenience they caused in your reading. The manuscript has been thoroughly revised and edited by a native speaker, so we hope it can meet the journal's standards. Thanks so much for your useful comments.

Q8: In the introduction, I would suggest including some references to the dielectric characterization of breast tissues and some references in the field of breast cancer diagnostics. This is to give a better picture of the situation in the fight against breast cancer.

We gratefully thanks for the precious time the reviewer spent making constructive remarks. We have added some references about breast cancer diagnostics. Because breast cancer is diagnosed by medical imaging, we have added the sentence ‘Breast cancer can be diagnosed by medical imaging including ultrasound[4, 5], mammography[6], and MRI[7].’. According to the type of medical imaging for breast cancer, the references Wang et al. [4] and Piotrzkowska et al. [5] for ultrasound, the reference Arita et al. [6] for mammography, and the reference Li et al. [7] for MRI are cited in the Introduction section.

Round 2

Reviewer 2 Report

Thank you very much to the authors for submitting the revised version of their work. I have just two more suggestions:

- Referring to equation (4), I recommend also defining what En fields are.

- Could you report also in English the output of the power amplifiers (in figure 8)?

Author Response

Thank you for your precious comments and advice. Those comments are all valuable and very helpful for revising and improving our paper, as well as the important guiding significance to our research. We have studied the comments carefully and have made corrections which we hope meet with approval. Revised portions are marked in red on the paper. The main corrections in the paper and the responses to the reviewer’s comments are as flowing:

Q1: Referring to equation (4), I recommend also defining what En fields are.

Thank you for pointing out this problem in the manuscript and we deeply appreciate the reviewer’s suggestion. We have changed the sentence ‘where φn and an present the input phases and amplitudes of array elements, respectively, and N is the number of antennas. Focused treatment is improved by optimizing the phases and amplitudes of antenna elements[31].’ to ‘In Equation (4), En is the electric field generated by unit power and zero-phase excitation of the nth antenna, N is the number of antennas, and φn and an present the input phases and amplitudes of array elements, respectively.’

Q2: Could you report also in English the output of the power amplifiers (in figure 8)?

Our deepest gratitude goes to you for your careful work and thoughtful suggestions that have helped improve this paper substantially. The output power of the power amplifier is consistent with the input power of the antenna. When the excitation of the antenna element is optimized using the algorithm, the radiated power of the antenna element is calculated from the phase and amplitude of the antenna element. The input power of the antenna is calculated by the radiated power and antenna efficiency. Therefore, we have added the sentence ‘An optimization algorithm technique is used to determine the optimal power and phase settings of the phased array antenna to direct power deposition at the tumor position. Hence, the output power of the power amplifier is derived from the radiated power of the antenna calculated from the optimized phase and amplitude of the antenna element for selected power deposition at the breast cancer position.’